# Microbiological and Cytokine Profiling of Menstrual Blood for the Assessment of Endometrial Receptivity: A Pilot Study

**DOI:** 10.3390/biomedicines11051284

**Published:** 2023-04-26

**Authors:** Mark Jain, Elena Mladova, Anna Shichanina, Karina Kirillova, Anna Povarova, Liya Scherbakova, Larisa Samokhodskaya, Olga Panina

**Affiliations:** 1Medical Research and Educational Center, Lomonosov Moscow State University, 119992 Moscow, Russia; 2Faculty of Medicine, Lomonosov Moscow State University, 119991 Moscow, Russia; 3Institute of Reproductive Medicine “REMEDI”, 123100 Moscow, Russia

**Keywords:** endometrial receptivity, cytokines, chemokines, growth factors, cytokine profile, menstrual blood, endometrial microbiota, assisted reproduction, in vitro fertilization

## Abstract

Endometrial receptivity (ER) is a key factor required for the successful implantation of the embryo. However, the evaluation of ER is challenging, as a nondisruptive sampling of endometrial biomaterial by conventional methods is only possible outside of the embryo transfer (ET) cycle. We propose a novel approach for the assessment of ER—microbiological and cytokine profiling of menstrual blood aspirated directly from the uterine cavity at the beginning of the cryo-ET cycle. The aim of the pilot study was to evaluate its prognostic potential regarding the outcome of the in vitro fertilization procedure. Samples collected from a cohort of 42 patients undergoing cryo-ET were analyzed by a multiplex immunoassay (48 various cytokines, chemokines, and growth factors) and a real-time PCR assay (28 relevant microbial taxa and 3 members of the *Herpesviridae* family). Significant differences between groups of patients who achieved and did not achieve pregnancy were observed for G-CSF, GRO-α, IL-6, IL-9, MCP-1, M-CSF, SDF-1α, TNF-β, TRAIL, SCF, IP-10, and MIG (*p* < 0.05), whereas microbial profiles were not associated with the outcome of cryo-ET. It appeared that levels of IP-10 and SCGF-β were significantly lower (*p* < 0.05), in patients with endometriosis. Menstrual blood may provide great opportunities to noninvasively investigate various parameters of the endometrium.

## 1. Introduction

Infertility is a disorder of the reproductive system defined by an inability to achieve pregnancy for at least 12 months of regular unprotected intercourse [1]. Every year, approximately 15% of couples of reproductive age are affected by this condition worldwide [2]. It is estimated that the female factor accounts for 50% of these cases [3].

Currently, assisted reproductive technologies (ART), such as in vitro fertilization (IVF) and intracytoplasmic sperm injection (ICSI), are key tools for the treatment of infertility—millions of respective procedures are performed every year. However, only 25% of them lead to successful delivery [4]. Given the high cost of these procedures and the possible adverse effects of repeated stimulated cycles, the development of reliable approaches for the prediction of the ART treatment outcome is among the main challenges of modern reproductive medicine [5].

There are two major factors responsible for implantation failure—embryo quality and endometrial receptivity (ER). It is estimated that their contribution to the outcome of ART procedures is approximately 30% and 70%, respectively [6,7]. The implementation of morphological grading and preimplantation genetic testing allowed to control the embryo quality and diminish the negative impact of this factor at least to some extent, though there are still no reliable options to evaluate ER prior to the embryo transfer (ET) [8,9].

ER is defined as the ability of the endometrium to provide the embryo with the opportunity to attach, invade, and develop, securing optimal microenvironmental and trophic statuses [10]. There is growing evidence that this embryo–endometrial crosstalk is likely mediated not only by certain mucosal signaling molecules [11] but also by the endometrial microbiota [12,13]. Various approaches based on ultrasonographic analysis were proposed to evaluate ER noninvasively, though, unfortunately, they are characterized by quite-low predictive value [14], whereas direct sampling of the endometrium appears to be challenging.

On the one hand, an endometrial biopsy provides sufficient material for any kind of analysis, though it is a highly invasive procedure, which is not allowed in the ET cycle (and cycle-to-cycle consistency of certain results is not guaranteed). On the other hand, some authors propose aspiration of endometrial fluid right before the ET and/or collection of ET cannula leftover cells and mucus, which are great for getting a glimpse of the ER at the most important point of the cycle, but have little to no prognostic value, as there are no clinical decisions to make after the event of ET [15,16].

It appears that menstrual blood is devoid of the above-mentioned shortcomings but is somehow largely ignored as a substrate for the assessment of ER. First, this biomaterial is in continuous contact with the endometrium and may contain any relevant molecular/microbial components of the tissue. Second, as it originates from the spiral arteries, it may reflect the status of the local microcirculation system. Third, menstrual blood may be easily and painlessly sampled directly from the uterine cavity, as the cervix is dilated during menstruation. This approach allows for reducing contamination by the contents of the lower reproductive tract and by the manipulations associated with the handling of menstrual cups. Lastly, this biomaterial may be collected at the beginning of the ET cycle, potentially providing an opportunity to reschedule the implantation based on the ER analysis findings.

The aim of the present pilot study was to evaluate the usefulness of menstrual blood as a biomaterial for the assessment of ER by comparing its microbiological and cytokine profiles in patients with different outcomes of ART treatment.

## 2. Materials and Methods

### 2.1. General Information

The study was approved by the institution’s Local Ethics Committee (#12/20, 21 December 2020) and conducted according to the tenets of the Declaration of Helsinki. Patients were enrolled from January 2021 to October 2022. All participants provided signed informed consent forms. The study included 42 patients scheduled for cryo-ET. No exclusion criteria were applied, so that the study cohort reflects the general population of an assisted reproduction clinic. All ART procedures were done strictly in concordance with the latest version of national clinical recommendations (ET was performed for blastocysts of 2BB and higher according to the Gardner grading system) [17]. Patients were divided into two groups based on the outcome of ET. Clinical pregnancy was verified by human chorionic gonadotropin testing with subsequent ultrasonographic visualization of one or more gestational sacs. In all cases of successful ET outcomes, the pregnancy was maintained for at least 12 weeks. The live-birth rate was not evaluated for the purposes of the present pilot study. The detailed clinical and demographic characteristics of the study participants are presented in Table 1, whereas all relevant depersonalized data regarding each enrolled patient are available in Appendix A.

### 2.2. Biomaterial Collection and Processing

Menstrual blood was collected on days 2–3 of the cryo-ET cycle directly from the uterine cavity using an ET catheter Guardia Access Transfer Catheter K-JETS-7019 (Cook Medical Inc., Bloomington, IN, USA) connected to a syringe. According to the feedback of the study participants, this method appeared to be painless. External genitalia as well as the vagina and cervix were disinfected with a Octenisept spray (Schulke & Mayr GmbH, Norderstedt, Germany) prior to the introduction of the ET catheter. Vaginal secretions were cleaned by cotton buds. The amount of aspirated menstrual blood was varying, in most cases it was not possible to sample more than 0.5 mL. Due to the high viscosity of the biomaterial, the catheter was often clogged and had to be rinsed with 0.2 mL of sterile (0.9%, NaCl) saline solution.

The menstrual blood samples were centrifuged at 5500 rpm for 10 min. The menstrual supernatant was transferred to separate tubes. Both menstrual sediment and supernatant were stored at −80 °C until further analysis.

### 2.3. Analysis of Immune Mediators

After defrosting, the menstrual supernatant was diluted 1:1 with a sterile (0.9%, NaCl) saline solution. The analysis of immune factors was carried out using a magnetic bead-based multiplex immunoassay Bio-Plex Pro™ Human Cytokine Screening 48-Plex on a Bio-Plex 200 instrument (Bio-Rad Laboratories Inc., Hercules, CA, USA). The panel included the following analytes: fibroblast growth factor (FGF) basic; Eotaxin; granulocyte colony-stimulating factor (G-CSF); granulocyte-macrophage colony-stimulating factor (GM-CSF); interferon (IFN) γ; IFN-α2; interleukin (IL) 1β; IL-1ra; IL-1α; IL-2Rα; IL-3; IL-12 (p40); IL-16; IL-2; IL-4; IL-5; IL-6; IL-7; IL-8; IL-9; IL-10; IL-12 (p70); IL-13; IL-15; IL-17A; IL-18; growth regulated oncogene (GRO) α; hepatocyte growth factor (HGF); leukemia inhibitory factor (LIF); monocyte chemotactic protein (MCP) 1; MCP-3; IFN-γ-induced protein (IP) 10; monokine induced by IFN-γ (MIG); β nerve growth factor (NGF); stem cell factor (SCF); stem cell growth factor (SCGF) β; stromal cell-derived factor (SDF) 1α; macrophage inflammatory protein (MIP) 1α; MIP-1β; platelet-derived growth factor (PDGF-BB); regulated upon activation normal T cell expressed and presumably secreted (RANTES); tumor necrosis factor (TNF) α; TNF-β; vascular endothelial growth factor (VEGF); cutaneous T cell-attracting chemokine (CTACK); macrophage migration inhibitory factor (MIF); TNF-related apoptosis-inducing ligand (TRAIL); macrophage colony-stimulating factor (M-CSF). Each sample was measured in duplicate.

Due to the inconsistency of menstrual blood sampling (varying volume, an unknown proportion of blood to endometrial fluid, etc.), the levels of all analyzed immune factors were normalized by total protein and expressed as pg per mg of total protein. An automated AU480 Chemistry Analyzer (Beckman Coulter, Brea, CA, USA) was used to measure the total protein in the menstrual supernatant. Approximately half of the samples exhibited visual signs of hemolysis, and hemoglobin is known to interfere with fluorescence-based immunoassays. Therefore, hemoglobin level was analyzed spectrophotometrically using a Multiskan GO instrument (Thermo Fisher Scientific Inc., Waltham, MA, USA) to evaluate its impact on the measurements of immune mediators.

### 2.4. Microbiological Analysis

After defrosting, menstrual sediment was resuspended in 0.2 mL of sterile (0.9%, NaCl) saline solution. DNA was isolated from 0.4 mL of biomaterial using QIAamp DNA Mini Kit (Qiagen GmbH, Hilden, Germany). The microbiological profiles were analyzed using the following reagent kits on a DT-Prime real-time PCR instrument (DNA-Technology, Moscow, Russia) according to the manufacturers’ protocols: ‘Femoflor 16′, ‘TNC Complex’, ‘Herpes Multiplex’ (DNA-Technology, Moscow, Russia). These reagents allowed quantitative analysis of the total bacterial load (based on the detection of conservative procaryotic sequences), *Lactobacillus* spp., *Enterobacteriaceae*, *Streptococcus* spp., *Staphylococcus* spp., *Gardnerella vaginalis*, *Prevotella bivia*, *Porphyromonas* spp., *Eubacterium* spp., *Sneathia* spp., *Leptotrichia* spp., *Fusobacterium* spp., *Megasphaera* spp., *Veillonella* spp., *Dialister* spp., *Lachnobacterium* spp., *Clostridium* spp., *Mobiluncus* spp., *Corynebacterium* spp., *Peptostreptococcus* spp., *Atopobium vaginae*, *Candida* spp., *Mycoplasma hominis*, *Ureaplasma urealyticum*, *Ureaplasma parvum*, and *Mycoplasma genitalium* as well as qualitative analysis of *Trichomonas vaginalis*, *Neisseria gonorrhoeae*, *Chlamydia trachomatis*, *Herpes simplex viruses 1 & 2*, and *Cytomegalovirus*. The amount of human DNA was also measured in every sample to control the biomaterial collection quality (>10^3^ copies per reaction mixture). Real-time PCR data was analyzed automatically in the RealTime_PCR software, version 7.9 (DNA-Technology, Moscow Russia) developed for the above-mentioned PCR kits. The recommended-by-the-manufacturer cycle-threshold value for qualitative analysis was set at 24. Results of the quantitative analysis were normalized by the total bacterial load and presented as ‘abundance’ (%). In order to verify the absence of sample contamination at various stages of the analysis, negative controls (sterile 0.9% saline solutions) were subjected to all stages of sample preparation and analysis, including exposure to the same ET catheters and tubes, isolation of DNA, and further real-time PCR.

### 2.5. Statistical Analysis

Data were analyzed using IBM SPSS Statistics 26.0 Software (IBM Corp., Armonk, New York, NY, USA). Data distribution normality was evaluated using Shapiro–Wilk’s test. Due to the absence of normal distribution for all tested variables nonparametrical statistical tests were used. Data are presented as median [interquartile range]. Qualitative data were compared using Fisher’s exact test, whereas the quantitative data were compared using a Mann–Whitney U test. The correlation of quantitative variables was assessed via Spearman’s rank coefficient (r_S_), which was interpreted using Chaddock’s scale. Shannon’s indexes were calculated to describe the α-diversity of the microbiological communities. Note that due to limitations of the selected real-time PCR microbiological panel, Shannon’s index is only reflecting the diversity within bacterial vaginosis-associated microbiota boundaries and should not be directly compared to other α-diversity data obtained in 16S rRNA sequencing studies. A *p*-value of less than 0.05 was considered statistically significant. Due to the exploratory design of the study and the variables being mostly dependent the need for an application of any correction for multiple comparisons is debatable and may sufficiently increase the likelihood of type II errors (especially given the large number of analyzed parameters) [18,19]. Yet, we decided to present the data in both ways. To maintain the visual clarity, we did not adjust the *p*-values themselves (e.g., Bonferroni correction), but controlled the false discovery rate (FDR) using Benjamini-Hochberg adjustment at FDR of 0.2. Therefore, throughout the text, the *p*-values (or the respective variables) that successfully passed the threshold for the correction for multiple comparisons of groups are marked with an asterisk “*”.

## 3. Results

### 3.1. Cytokine Profiling

All 48 immune mediators included in the panel were detectable in the menstrual supernatant. Cases, when a given sample had a level of a cytokine below the detection limit, were rare (87 out of 2016 analyses). A heat map representing the results in each individual sample is presented in Figure 1, whereas numerical data are available in Appendix A.

Among the analyzed immune factors, statistically significant differences between groups of patients who achieved and did not achieve pregnancy were observed for G-CSF, GRO-α, IL-6, IL-9, MCP-1, M-CSF *, SDF-1α, TNF-β, TRAIL *, SCF *, IP-10 *, and MIG * (*p* < 0.05) (Figure 2). Median [interquartile range] levels of each studied cytokine for the total cohort and both groups (with corresponding *p*-values) are presented in Appendix A. It is worth noting that IP-10 and MIG exhibited significant prognostic value regarding the outcome of the ART procedure (areas under the ROC curves of 0.762 and 0.773, respectively, Appendix A). At a cutoff level of 17.99 pg/mg, IP-10 was allowed to predict pregnancy with a sensitivity of 95%, and a specificity of 57%. In other words, patients with levels of IP-10 below the above-mentioned cutoff value had a chance of achieving pregnancy as low as 5%, although these results must be validated in a larger cohort.

The broad range of studied analytical targets allowed us to get an insight into the interplay between immune mediators in the menstrual blood, which was visualized as a correlation matrix (Figure 3). It appeared that all 12 cytokines, which were upregulated in patients who achieved pregnancy, correlated with each other significantly with rare exceptions. The correlation was low-to-high positive (r_S_ ranging from 0.31 to 0.77 for statistically significant correlations).

Among the relevant clinical parameters, only endometriosis and diminished ovarian reserve (DOR) were sufficiently prevalent in our cohort for the comparison of immune mediators. It appeared that the levels of IP-10 and SCGF-β were significantly lower in the menstrual supernatant of patients with endometriosis compared to those who did not have this condition (16.03 [7.12; 33.57] vs. 29.23 [17.91; 65.38] pg/mg, *p* = 0.044 and 3937 [2499; 5989] vs. 7206 [3379; 12,073] pg/mg, *p* = 0.015 *, respectively), whereas samples of patients with DOR exhibited lower levels of IL-10 and MIG (0.66 [0.35; 1.20] vs. 1.46 [1.03; 2.49] pg/mg, *p* = 0.006 * and 6.88 [4.04; 10.79] vs. 14.30 [6.77; 29.70] pg/mg, *p* = 0.008 *, respectively) (Figure 4). Age did not correlate significantly with either analyzed immune mediator (*p* > 0.05), whereas body-mass index did with CTACK, IL-7 (negative moderate effect: r_S_ of −0.58 and −0.51, respectively; *p* < 0.001), and SDF-1α (negative low effect: r_S_ of −0.35; *p* < 0.05) (Appendix A).

Unexpectedly, the presence of hemolysis in the sample had a minor impact on the measurements of cytokines. Statistically significant (*p* < 0.05) correlation with hemoglobin was observed only for Eotaxin, GM-CSF, IL-4, IL-9, IL-10, IL-18, MIP-1β, β-NGF, SCF, SDF-1α, TNF-β (positive low effect: r_S_ of 0.31–0.45), CTACK, PDGF-BB, and RANTES (positive moderate effect: r_S_ of 0.53–0.65). Therefore, the levels of hemoglobin were not used for any normalization purposes. Detailed results of the correlation analysis for hemoglobin and all the studied immune mediators are available in Appendix A.

### 3.2. Microbiological Profiling

All samples successfully passed the quality control test and exhibited the presence of human DNA (5.0 × 10^5^ [1.6 × 10^5^; 1.3 × 10^6^] copies per reaction mixture). Total bacterial DNA was detectable in all samples as well, although at significantly lower levels (2.3 × 10^3^ [0.6 × 10^3^; 7.9 × 10^3^] copies per reaction mixture). There was no correlation between the total bacterial DNA load and the level of human DNA (r_S_ = 0.07, *p* = 0.645). No contamination by microbial and viral (at least regarding the analyzed targets) DNA was found in the negative control samples.

The results of the microbiological profiling of the menstrual sediment are summarized in Figure 5; complete numerical values are available in Appendix A. Abundances and detection rates of the analyzed taxa, as well as total bacterial DNA (absolute values and ratios to human DNA) did not differ significantly in the comparison of patients based on the outcome of ART treatment, the presence of endometriosis and DOR (*p* > 0.05). Boxplots for the comparisons of the abundances for taxa with near-significant *p* values (e.g., 0.063–0.059) are available in Appendix A.

Analysis of the α-diversity of microbiota revealed that Shannon’s indexes were quite low in general (0 [0; 0.34]), which might be a result of a limited number of analyzed taxa and not the total microbiota composition of the biomaterial itself. There were no significant differences in the comparison of patients who achieved and did not achieve pregnancy (0 [0; 0.45] vs. 0 [0; 0.30], *p* = 0.401, respectively) (Appendix A). Shannon’s indexes did not differ based on the presence of endometriosis (*p* > 0.05), although they did slightly in patients with DOR (0 [0; 0] vs. 0.0002 [0; 0.6384], *p* = 0.02 *) (Appendix A). Among the analyzed microbiological parameters, age correlated significantly only with Shannon’s indexes (negative low effect: r_S_ of −0.40; *p* = 0.01), whereas no correlations were observed for body mass indexes (Appendix A).

Finally, we assessed the relations of various data generated during microbiological analysis with the levels of studied immune mediators. It appeared that the abundance of *Lactobacillus* spp. correlated with the levels of G-CSF (r_S_ of 0.36; *p* = 0.019) and M-CSF (r_S_ of 0.35; *p* = 0.022), whereas Shannon’s α-diversity indexes with RANTES (r_S_ of 0.38; *p* = 0.012). The correlation was positive with a low effect size in both cases (Appendix A).

## 4. Discussion

In the past decades, menstrual blood has been established as one of the main sources of mesenchymal stem cells for regenerative medicine [20]. However, this biomaterial has been largely overlooked for diagnostic/prognostic purposes. Available data regarding the analysis of any immune mediators in the menstrual blood are scarce.

In a recent study, Guterstam et al. analyzed a panel of 20 cytokines, chemokines, and growth factors in paired samples of menstrual (collected using a menstrual cup) and peripheral blood in 19 asymptomatic volunteers [21]. The authors observed significant differences in the concentrations of all tested immune mediators, with an exception for the levels of osteopontin; the most pronounced differences were demonstrated for IL-6, IL-1β, and IL-8 (in most cases, menstrual blood had higher levels of the analyte), indicating that this biomaterial is indeed reflective of the mucosal biomarkers of the endometrium. The patterns of cytokines matching those selected for our study on the heat map provided by Guterstam et al. slightly differed from our results, which might be explained by the differences in the sampling process (contamination of the biomaterial by the content of mucosa in the lower parts of the reproductive tract on its way to the menstrual cup, the normalization of cytokines levels by total protein in our study, etc.).

Tortorella et al. utilized a sampling approach similar to ours and collected the menstrual blood directly from the uterine cavity for the analysis of three common proinflammatory cytokines (IL-1β, IL-6, and TNF-α) to develop a new noninvasive approach to diagnose chronic endometritis (CE) [22]. It is worth noting that the authors did not normalize the levels of cytokines and reported the volume of collected biomaterial >0.5 mL. In our experience, the sampling of this volume of menstrual blood was rarely possible, which was among the reasons for us to normalize the data by total protein. ROC analysis revealed that the IL-6/TNF-α-based model was an extremely efficient predictor of CE (AUC of 0.989). Unfortunately, we did not perform an endometrial biopsy for the diagnosis of CE based on histological findings to compare the results. It is important to mention that according to the recent meta-analysis, CE significantly reduces the clinical pregnancy rate in women undergoing IVF (odds ratio 2.28, *p* = 0.002), whereas CE cure drastically increases this parameter (odds ratio 3.64, *p* = 0.0001) [23]. Therefore, menstrual blood may be of great value not only for the assessment of ER itself but also for the detection and monitoring of CE.

In our cohort, we observed a pattern of proinflammatory immune mediators and growth factors related to the stem cells (G-CSF, GRO-α, IL-6, IL-9, MCP-1, M-CSF *, TNF-β, TRAIL *, IP-10 *, MIG * and SDF-1α, SCF *, respectively) being slightly upregulated in patients who achieved pregnancy. This does not by any means indicate an active inflammation in this group, but rather a proinflammatory phenotype with an increased proliferative status, which is a commonly accepted concept of the receptive endometrium [24,25].

There are several studies available, which were dedicated to the analysis of 2–17 cytokines in the endometrial fluid aspirated on the day of ET [16,26,27]. They were contradicting each other in terms of the association of IL-1β and TNF-α with ER, although supporting our results regarding the upregulated levels of IP-10 * and MCP-1 in cases of successful implantation, indicating that the proinflammatory status of the receptive endometrium might be present throughout the entire menstrual cycle. Our results regarding the possible downregulation of IP-10 in the endometria of patients with endometriosis are to some extent concordant with the data obtained in endometrial biopsy and peritoneal fluid specimens, although this biomarker was characterized by borderline significance [28,29]. Therefore, IP-10 might be mediating endometriosis, though its relevancy for any diagnostic/prognostic purposes is limited.

For decades, the endometrium was considered to be a sterile environment [30,31]. In recent years due to the advent of sensitive microbiological techniques, this concept is being actively disputed [32]. Transcervical sampling is in most cases the only available option for the studies of endometrial microbiota, especially in patients of ART clinics, whereas there is no proper way to entirely exclude the contamination of samples by microbes from the lower parts of the reproductive tract. There are several studies demonstrating that endometrial samples obtained at hysterectomy via surgical access still contained bacterial genetic material, although the detection rate of *Lactobacillus* spp. was quite low 0–23% [33,34,35]. However, there are reports with similar sampling approaches and contradictory results, indicating that *Lactobacillus* spp. is abundant, and the overall profile of microbiota is similar to that in biomaterial obtained transcervically [36,37].

Overall, the impact of endometrial microbiota on ER is still unknown, and the results of available studies in this field are inconsistent [32,38,39]. Unfortunately, due to severe methodological heterogeneity, it is difficult to conduct a proper meta-analysis to summarize the published data. We did not observe any statistically significant connections between the analyzed parameters of microbiota and the outcome of ART treatment. Three patients in our cohort had near 100% abundance of either *Enterobacteriaceae* or *Candida* spp. (Figure 5), which can hardly be classified as a healthy microbial community. Remarkably, none of them achieved pregnancy. These cases failed to reach statistical significance, although may become relevant upon extension of the cohort.

The observed positive correlation of *Lactobacillus* spp. with G-CSF and M-CSF might give a hint on the possible mechanism by which normal microbiota modulates ER (assuming that this taxon really resembles normal microbiota [13]). It is known that both the above-mentioned cytokines may enhance endometrium growth and maturation, which is crucial for successful implantation [40,41,42]. Moreover, in our cohort, these immune mediators were slightly upregulated in patients who achieved pregnancy.

This pilot study had certain limitations. First, the study groups were small. Therefore, it was not possible to create an unbiased prognostic model including multiple variables. Second, we did not use any approaches to reduce the chances of contamination of menstrual blood by cervical microbiota, such as the introduction of an external sheath to the sampling catheter. This was done to keep the procedure painless. Third, it is worth noting that cytokine profiles at the beginning of the menstrual cycle may not reflect those on the day of ET, although this fact may not necessarily influence the presented results as the statistical analysis was based on the outcome of cryo-ET. Last, due to the exploratory design of the study the criteria for the multiple comparison adjustment were not strict. The study aimed to narrow the list of potential biomarkers for future studies on this biomaterial and highlight the most promising targets, rather than ultimately establish biomarkers associated with ER.

## 5. Conclusions

Menstrual blood may provide great opportunities to noninvasively investigate various parameters of the endometrium at the beginning of the ET cycle. From the clinical perspective, this timing is ideal to assess the ER and potentially reschedule the ART procedure. In the present pilot study, we provided the most comprehensive molecular and microbiological characterization of this biomaterial sampled directly from the uterine cavity to date. Future studies involving menstrual blood as a biological substrate for the assessment of ER with larger cohorts of patients are anticipated. The discovery of a reliable approach to evaluate this elusive and complex parameter of the endometrium is crucial not only to predict the outcome of ART treatment but also to determine potential therapeutic targets to influence it.

## Figures and Tables

**Figure 1 biomedicines-11-01284-f001:**
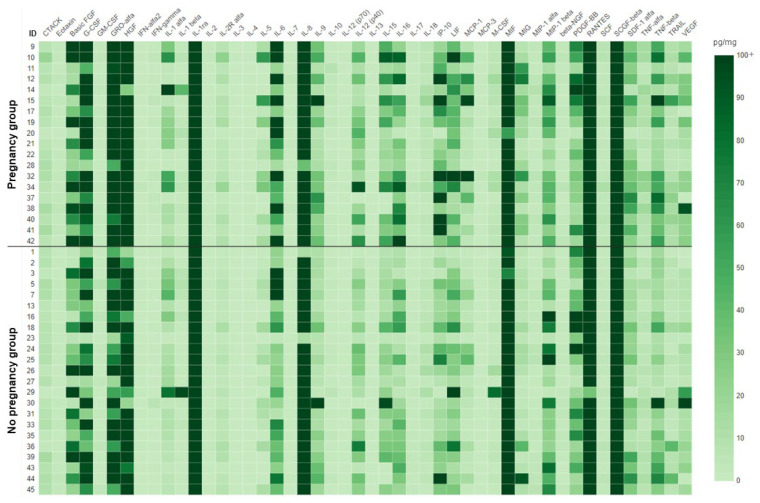
Heat map for the cytokine analysis in menstrual supernatant. Data presented as pg per mg of total protein. The scale was capped at 100 pg/mg to preserve the color separation at lower concentrations of cytokines. Uncapped numerical values for each sample are available in Appendix A.

**Figure 2 biomedicines-11-01284-f002:**
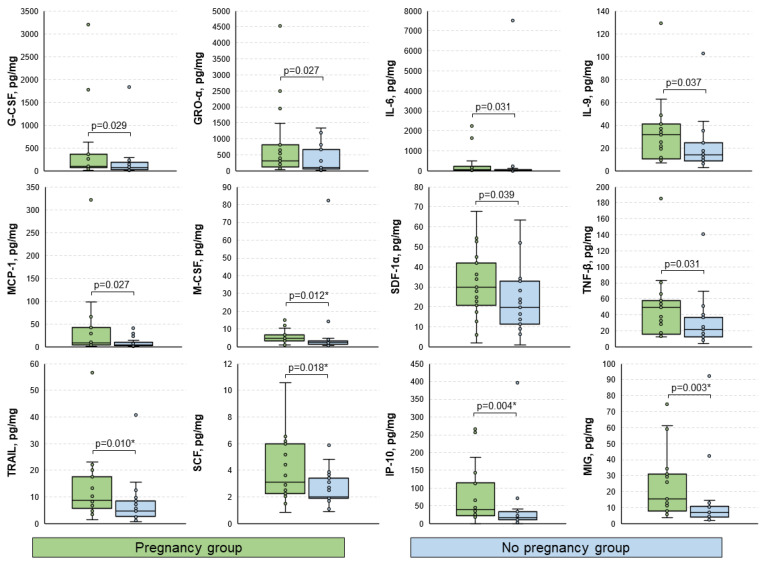
Boxplots for the comparison of levels of analyzed immune mediators between groups. Data presented as pg of analyte per mg of total protein. Only statistically significantly differing cytokines are shown. *p*-values were calculated using the Mann–Whitney U test. *, *p*-values significant after the Benjamini–Hochberg adjustment for multiple comparisons.

**Figure 3 biomedicines-11-01284-f003:**
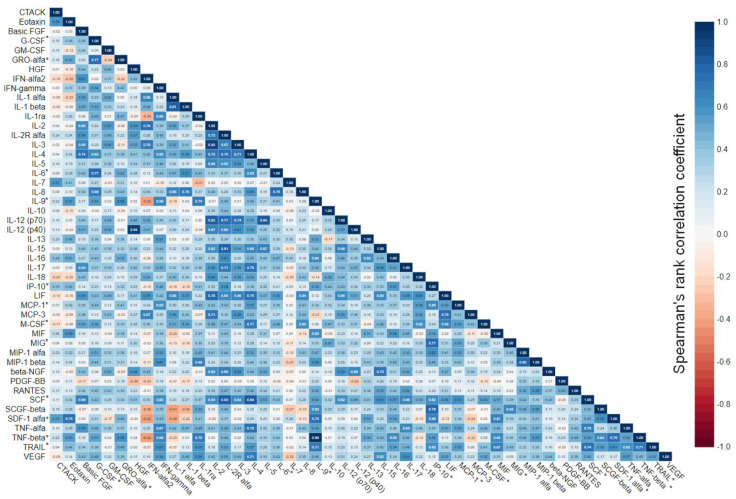
Correlation matrix for 48 immune mediators analyzed in menstrual supernatant. Correlation of quantitative variables was assessed via Spearman’s rank coefficient. Data were calculated based on the results in the total cohort of patients. Cytokines that were upregulated in patients who achieved pregnancy are marked with “*”.

**Figure 4 biomedicines-11-01284-f004:**
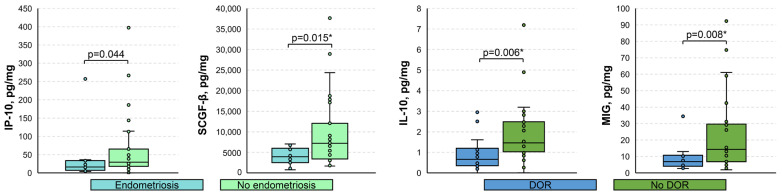
Boxplots for the comparison of levels of analyzed immune mediators based on the presence of endometriosis and diminished ovarian reserve. DOR, diminished ovarian reserve. Data presented as pg of analyte per mg of total protein. Only statistically significantly differing cytokines are shown. *p*-values were calculated using the Mann–Whitney U test. *, *p*-values significant after the Benjamini–Hochberg adjustment for multiple comparisons.

**Figure 5 biomedicines-11-01284-f005:**
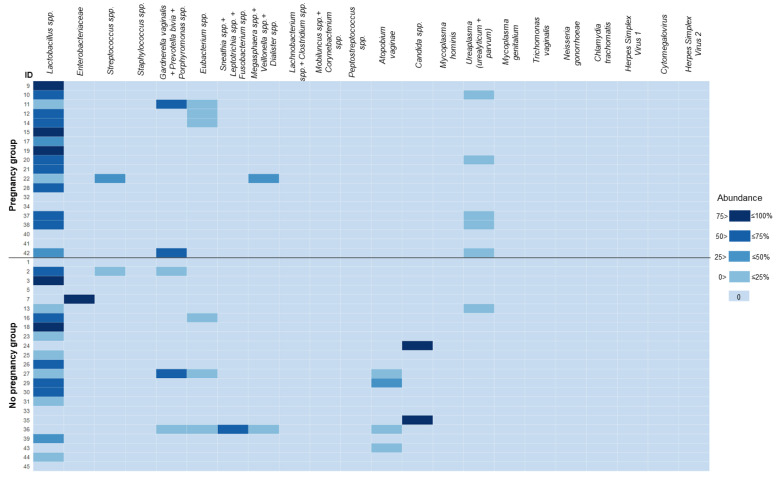
Heat map for the microbiological analysis in menstrual sediment. Data presented as abundance (% of total bacterial load measured by amplification of conservative prokaryotic DNA sequence). For convenience, the levels of *Candida* spp. DNA were added to the values of the total bacterial load. The heat-map scale was segmented to ensure a clear color separation for cells with values of 0%. Some closely related species/groups of species were analyzed collectively due to limitations of the applied real-time PCR assay. In such cases, the names of the taxa are listed together and joined by a “+” sign.

**Table 1 biomedicines-11-01284-t001:** The demographic and clinical characteristics of study participants.

Parameters	Pregnancy Group (*n* = 19)	No Pregnancy Group (*n* = 23)
Age, years	34.7 (19.0–50.0)	38.3 (31.0–51.0)
Body mass index, kg/m^2^	21.7 (17.3–30.5)	21.9 (17.3–27.9)
ART details: - IVF - IVF/ICSI - cryo-ET - stimulated cycle - natural cycle - PGT	12/19 7/19 19/19 18/19 1/19 6/19	9/23 14/23 23/23 23/23 0/23 13/23
Infertility factors: - tubo-peritoneal factor - endometriosis - DOR - uterine factor - endocrinologic factor - male factor - unknown factor	5/19 3/19 4/19 1/19 1/19 6/19 2/19	3/23 8/23 12/23 1/23 0/23 5/23 3/23
Uterine adhesions	0/19	1/23
Uterine fibroids	0/19	2/23
RIF	1/19	5/23
RPL	0/19	1/23

Data are presented as mean (range) or number observed/number of subjects. ART, assisted reproductive technologies; IVF, in vitro fertilization; ICSI, intracytoplasmic sperm injection; ET, embryo transfer; PGT, preimplantation genetic testing; DOR, diminished ovarian reserve; RIF, recurrent implantation failure; RPL, recurrent pregnancy loss. RIF and RPL were defined as ≥3 corresponding events in anamnesis.

## Data Availability

The data presented in this study are available in Appendix A.

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
