# Peer review of "Microbiological and Cytokine Profiling of Menstrual Blood for the Assessment of Endometrial Receptivity: A Pilot Study"

_biomedicines, 2023, doi:10.3390/biomedicines11051284_

Round 1
Reviewer 1 Report
This study examined whether microbiological and cytokine profiling of menstrual blood in the beginning of the cryo-ET cycle could be used as a non-invasive method for the prognosis of the outcome of the in vitro fertilization procedure. This study shows that menstrual blood provides great opportunities to non-invasively investigate various parameters of the endometrium. This study is very interesting considering non-disruptive sampling of endometrial biomaterial is impossible during the embryo transfer period. However, this study it has a critical concern in concluding this result.
1. Various parameters of endometrium including cytokine profiling are highly dependent on the phase of the menstrual cycle. Therefore, the cytokine profiling of endometrial cells remaining at the onset of menstruation cannot be considered the same as that at the time of embryo transfer. A rationale for this issue is required.
2. A larger number of samples will be required to ensure cycle to cycle consistency of certain results. Samples collected from a cohort of 42 patients seem insufficient to conclude this result.
Author Response
Comment 1: “Various parameters of endometrium including cytokine profiling are highly dependent on the phase of the menstrual cycle. Therefore, the cytokine profiling of endometrial cells remaining at the onset of menstruation cannot be considered the same as that at the time of embryo transfer. A rationale for this issue is required.”.
Response: We are grateful for this comment. Our team completely agrees that cytokine profiles of endometrial cells at the onset of menstruation must be different throughout the cycle. Moreover, we profiled not the endometrial cells themselves but the menstrual blood which brings all the cytokines from the peripheral blood into the equation. However, the goal of the cytokine profiling of menstrual blood is not to reflect the levels of immune mediators on the day of embryo transfer (which is probably impossible), but to reflect (predict) the endometrial receptivity on the day of embryo transfer within the same cycle. In this pilot study we based our main calculations on the outcome of frozen embryo transfer. Unfortunately, there are no published studies available in which the authors would analyze equally large panel of cytokines in the endometrial biomaterial on the day of embryo transfer, although there are some studies with smaller panels. These studies are mentioned in the discussion section and their results support our findings regarding the upregulation of IP-10 and MCP-1 in the receptive endometrium. The main drawback of the analysis of endometrial aspirates directly prior to the embryo transfer is the fact that there are no potential clinical decisions to make at that point.
Anyway, we have added the following lines to the study limitations paragraph (last paragraph of the discussion section) to make this point clear for the readers:
“…Thirdly, it is worth noting that cytokine profiles in the beginning of the menstrual cycle may not reflect those on the day of ET, although this fact may not necessarily influence the presented results as the statistical analysis was based on the outcome of cryo-ET…”
Comment 2: “A larger number of samples will be required to ensure cycle to cycle consistency of certain results. Samples collected from a cohort of 42 patients seem insufficient to conclude this result”.
Response: Thank you for pointing this out. The main goal of this pilot study was to bring attention of the scientific community to the possibility of the assessment of various molecular and microbiological markers in the menstrual blood in the beginning of the embryo transfer cycle. In the above-mentioned study limitations paragraph, we have pointed out the issue of small groups of patients and stated that our results are meant to narrow the list of potential biomarkers for future studies on this biomaterial and highlight the most promising ones, rather than to ultimately claim that certain biomarker is associated with endometrial receptivity. However, after rereading the last sentence of the abstract and the conclusions section we realized that we may have overclaimed that “menstrual blood provides great opportunities…”. Therefore, we have modified some sentences to keep the conclusions more in line with the limitations of the study:
“Menstrual blood may provide great opportunities to non-invasively investigate various parameters of the endometrium in the beginning of the ET cycle…”
“…Future studies involving menstrual blood as a biological substrate for the assessment of ER with larger cohorts of patients are anticipated…”
Additionally, some minor corrections (typos, etc.) were done throughout the text.
Reviewer 2 Report
The work by Jain and colleagues is timely, interesting and well-written. The pilot nature of the study doesn't allow for very strong conclusions, but the authors do a good balance of both presenting all data and not over-selling the results.
Two main points drew my attention during review:
The first one is that the authors used qPCR for the microbiome profiles, meaning that they have, in some sense, absolute abundance of their microbes. However, for analysis, they choose to transform these data to proportions. The use of proportions in microbiome data is often a necessity of sequencing methods, but there is a lot to be gained by using quantitative measures. Given the nature of the samples, the data still has to be normalized somehow, e.g. against total human DNA per sample. Additionally, there is previous data suggesting that low microbial density of the endometrium per se is associated to increased fertility. This could be assessed as the ratio of total bacterial DNA to total human DNA, in this cohort.
The second main point is that binary clinical characteristics have been assessed against both immune and microbial markers, but demographic characteristics known to affect fertility and/or the female microbiome are not explored. All markers should also be correlated to e.g. age, BMI and parity. This could change the interpretation of fig. 4 quite a lot.
Some details are lacking in the methods:
Please be explicit about the inclusion and exclusion criteria for this cohort, e.g. age, language fluency, reproductive history
Please describe the negative controls in more detail. Are these simply saline solution? Were they exposed to the examining room or to a clean catheter in any way?
Other minor points:
In table 1, PGT of the pregnancy groups is displayed as total of 16, not 19. Is this correct?
Figures 1 and 3 might be easier to read if the data is clustered (author's choice)
Figure 3 would definitely benefit by having the cytokines that were differentially abundant between groups marked, e.g. by a * or in bold font.
Fig S3 is described as a boxplot, but looks like a barplot. Is this a mistake or just an artefact of the skewed data distribution?
Author Response
Comment 1: “The first one is that the authors used qPCR for the microbiome profiles, meaning that they have, in some sense, absolute abundance of their microbes. However, for analysis, they choose to transform these data to proportions. The use of proportions in microbiome data is often a necessity of sequencing methods, but there is a lot to be gained by using quantitative measures. Given the nature of the samples, the data still has to be normalized somehow, e.g. against total human DNA per sample. Additionally, there is previous data suggesting that low microbial density of the endometrium per se is associated to increased fertility. This could be assessed as the ratio of total bacterial DNA to total human DNA, in this cohort”.
Response: Thank you for this comment. We decided to normalize the absolute levels of each analyzed taxa against total bacterial load for several reasons. Firstly, we calculated the total bacterial load based on the amplification of highly conservative procaryotic sequence (which allows to estimate the total load of 95-98% of all bacterial genomes). Therefore, the bacterial abundances which we calculated are to some extent comparable to the abundances which are presented in various studies based on 16S rRNA sequencing. This fact allows to compare our data with the other published results and eventually it may become useful for authors conducting a meta-analysis. Secondly, to our mind, proportions (bacterial abundances) provide a better insight on the exact position of a certain taxon in the microbial community of the endometrium. Thirdly, normalization against human DNA in the case of menstrual blood (in contrast to endometrial biopsies / aspirates) may not be legitimate. It is hard to say what are the proportions of endometrial biomaterial to the cellular components of the blood, what is the consistency of this proportion from sample to sample (this was among the reasons why we normalized cytokine levels against total protein). Moreover, it is well known that the amount of circulating cell-free DNA may significantly differ between individuals, which may even more increase the discrepancy (and potentially skew the data) in the case of normalization using human DNA. Therefore, in the case of menstrual blood such approach to normalization of the data is probably not going to reflect the ratio of bacteria to endometrial cells. As far as we know, this study is the first to analyze microbiota in the menstrual blood sampled directly from the uterine cavity; and there is no third-party experience to rely on.
Anyway, the suggestion to account the human DNA is interesting. Unfortunately, we did not reveal any noteworthy changes in the significance of the carried out comparisons between groups when the data was normalized against human DNA. However, we decided to include the suggested topic of low microbial density (what is important is that its addition may not have such a detrimental effect on the issue of multiple comparisons, when compared to the inclusion of all microbiological data normalized against human DNA). The following lines were added/modified:
“Abundances and detection rates of the analyzed taxa, as well as total bacterial DNA (absolute values and ratios to human DNA) did not differ significantly in the comparison of patients based on the outcome of ART treatment, the presence of endometriosis and DOR (p>0.05).”
Comment 2: “The second main point is that binary clinical characteristics have been assessed against both immune and microbial markers, but demographic characteristics known to affect fertility and/or the female microbiome are not explored. All markers should also be correlated to e.g. age, BMI and parity. This could change the interpretation of fig. 4 quite a lot”
Response: Thank you for pointing this out. It is indeed our mistake to not present such valuable data. The following lines were added to the results section after the data regarding diminished ovarian reserve:
“Age did not correlate significantly with neither analyzed immune mediator (p>0.05), whereas body mass index did with CTACK, IL-7 (negative moderate effect: rS of -0.58 and -0.51, respectively; p<0.001), and SDF-1α (negative low effect: rS of -0.35; p<0.05) (Fig. S2).”
“Among the analyzed microbiological parameters, age correlated significantly only with Shannon’s indexes (negative low effect: rS of -0.40; p=0.01), whereas no correlations were observed for body mass indexes (Fig. S5).”
Respective figures were added to the supplementary materials. Parity was not assessed as this parameter equaled zero in most cases.
Comment 3: “Please be explicit about the inclusion and exclusion criteria for this cohort, e.g. age, language fluency, reproductive history”
Response: Thank you for this comment. We did not apply any exclusion criteria, as it was not needed considering the aim of the study (assess the endometrial receptivity by analyzing certain molecular and microbiological parameters of menstrual blood). There was even a patient of 51 years old who achieved pregnancy in our cohort. In other words, upon completion of the main study, we hope to create a tool which may allow to predict the outcome of the frozen embryo transfer based on the analysis of menstrual blood. For this purpose, the study cohort must resemble general population of an assisted reproduction clinic. The following line was added to the materials and methods section:
“The study included 42 patients scheduled for cryo-ET. No exclusion criteria were applied, so that the study cohort reflects general population of an assisted reproduction clinic”.
Comment 4: “Please describe the negative controls in more detail. Are these simply saline solution? Were they exposed to the examining room or to a clean catheter in any way?”
Response: Thank you for pointing this out. We have indeed poorly described the negative control samples and their handling (moreover, they were mentioned separately in different paragraphs). The respective paragraph in the materials and methods section was rewritten for clarity:
“In order to verify absence of sample contamination at various stages of the analysis, negative controls (sterile 0.9% saline solutions) were subjected to all stages of sample preparation and analysis, including exposure to the same ET catheters and tubes, isolation of DNA and further real-time PCR.”
Comment 5: “In table 1, PGT of the pregnancy groups is displayed as total of 16, not 19. Is this correct?”
Response: Thank you for spotting this typo!
Comment 6: “Figures 1 and 3 might be easier to read if the data is clustered (author's choice)”
Response: Thank you for this suggestion! We tried to cluster the data, to our mind, in these figures it did not help, but only extended the figure frame forcing to orient it vertically.
Comment 7: “Figure 3 would definitely benefit by having the cytokines that were differentially abundant between groups marked, e.g. by a * or in bold font.?”
Response: Thank you for pointing this out! Respective correction to the figure and its legend were made.
Comment 8: “Fig S3 is described as a boxplot, but looks like a barplot. Is this a mistake or just an artefact of the skewed data distribution?”
Response: It is a boxplot, but the median equals zero, therefore, it looks like a barplot.
Round 2
Reviewer 1 Report
Dear Authors
Your responses to my review are well documented and allow me to fully understand the results of your research. Thank you for your sincere reply.